# Voice Prosthesis Coated with Sustained Release Varnish Containing Clotrimazole Shows Long-Term Protection against *Candida albicans:* An In Vitro Study

**DOI:** 10.3390/molecules26175395

**Published:** 2021-09-05

**Authors:** Ronit Vogt Sionov, Irith Gati, David Kirmayer, Michael Friedman, Doron Steinberg, Menachem Gross

**Affiliations:** 1The Biofilm Research Laboratory, The Institute of Dental Sciences, The Faculty of Dental Medicine, The Hebrew University of Jerusalem, Jerusalem 9112102, Israel; dorons@ekmd.huji.ac.il; 2Institute for Drug Research, School of Pharmacy, The Hebrew University of Jerusalem, Jerusalem 9112102, Israel; irith.gati@mail.huji.ac.il (I.G.); davidki@ekmd.huji.ac.il (D.K.); michaelf@ekmd.huji.ac.il (M.F.); 3Department of Otolaryngology-Head and Neck Surgery, Hadassah Medical Center, Jerusalem 9112102, Israel; drgrossm@hotmail.com; 4The Faculty of Medicine, The Hebrew University of Jerusalem, Jerusalem 9112102, Israel

**Keywords:** biofilm, *Candida albicans*, clotrimazole, fungal infection, laryngeal malignancies, laryngectomy, sustained release varnish (SRV), voice prosthesis

## Abstract

Fungal biofilm formation on voice prosthesis (VP) is a major health problem that requires repeated replacement of the prosthesis. *Candida albicans* is one of the pathogens that frequently inhabits the VP. We proposed that coating VPs with sustained-release varnish (SRV) containing clotrimazole (CTZ) might prevent fungal biofilm formation. The long-term antifungal activities of SRV-CTZ- versus SRV-placebo-coated VPs was tested daily by measuring the inhibition zone of *C. albicans* seeded on agar plates or by measuring the fungal viability of *C. albicans* in suspension. The extent of biofilm formation on coated VPs was analyzed by confocal microscopy and scanning electron microscopy. We observed that SRV-CTZ-coated VPs formed a significant bacterial inhibition zone around the VPs and prevented the growth of *C. albicans* in suspension during the entire testing period of 60 days. Fungal biofilms were formed on placebo-coated VPs, while no significant biofilms were observed on SRV-CTZ-coated VPs. HPLC analysis shows that CTZ is continuously released during the whole test period of 60 days at a concentration above the minimal fungistatic concentration. In conclusion, coating VPs with an SRV-CTZ film is a potential effective method for prevention of fungal infections and biofilm formation on VPs.

## 1. Introduction

Voice prostheses (VPs) are indwelling silicone valves that are used for speech rehabilitation in patients with laryngeal malignancies after laryngectomy [1]. Microbial biofilm formation on VPs is a major problem that requires its frequent replacement and increases the risk of chest infection and pneumonia [2]. Microbial growth is promoted in the environment of the esophagus where they are exposed to food, saliva, liquids, humidity, and temperature close to 37 °C [1]. *Candida albicans* is the predominant fungal species and *Staphylococcus aureus* is the most frequent bacterial colonizer found on VPs [2]. In addition, *C. albicans* better colonizes surfaces in CO_2_-rich environments such as those provided by exhaled breath, and the fungi can also penetrate the silicone of the device, eventually rendering it completely unusable [1,2]. Due to fungal and bacterial proclivity for growth in such a favorable environment, the median VP lifespan is reported to range between 60–92 days, depending on the device [3,4].

A number of approaches have been proposed to increase the VPs lifespan, such as the incorporation of silver oxide (Blom Singer Advantage) and Teflon-like fluoroplastic (Provox ActiValve) within the VPs material to prevent microbial colonization [1,5,6]. Although the Blom Singer Advantage and Provox ActiValve showed reduced *Candida* biofilm formation in comparison to the unmodified Provox 2 and Provox Vega after repeated exposures to the fungi, still biofilms were formed on these devices [5]. Attempts have also been made to coat the silicone VPs with gold or titanium to create a *Candida*-resistant surface [7]. However, *Candida* biofilm was formed on the gold-coated VPs after 1 month use by the patients [7]. De Prijck et al. [8] covalently bound antimicrobial quaternary ammonium compounds dimethylamino-ethyl methacrylate and polyethylenimine to the silicon surfaces, and observed that it is possible to prevent *Candida* biofilm formation after one time exposure to the fungi for 24 h. Other approaches for decontamination of oropharyngeal yeast include amphotericin B lozenges and buccal adhesive slow-release tablets containing miconazole [1]. However, once biofilms are formed, it is extremely difficult to eradicate them since the microbes in biofilms are much less sensitive to antimicrobial drugs when compared to planktonic growth [9]. Hence, it is important to develop new strategies for the prevention of initial microbial colonization on the VPs. The prevention and control of biofilm formation is not only beneficial for the lifespan of the prosthesis, but also for overall patient health and for his quality of life.

Herein we propose a unique method for VP coating using a sustained-release varnish (SRV) containing the broad-spectrum antifungal drug clotrimazole (CTZ), that forms a thin film coating on the VPs. SRV is a pharmaceutical technology that has the advantage of delivering the drug in controlled dosages over a long period of time, thereby providing a more consistent antimicrobial environment [10]. Such varnishes embedded with different active agents have previously been shown to provide long-term antimicrobial activities in other medical systems including oral care [10] and catheter-associated urinary tract infections [11,12]. 

CTZ is a well-tolerated fungistatic drug which acts by targeting the biosynthesis of ergosterol, an essential component of the fungal plasma membrane [13]. In addition, CTZ and its related miconazole have been documented to exert antibacterial activity toward certain *Staphylococcus* species [14,15], although at 20–40 times higher concentrations than those needed for the fungistatic effect. In previous studies, a formulation of SRV-CTZ was demonstrated to relieve oral stomatitis and reduce the *Candida* level when applied to the upper denture [16]. Herein we demonstrate that this SRV may be utilized to protect voice prosthesis from *C. albicans* infection and biofilm formation, thereby extending the lifespan of the device.

## 2. Results

### 2.1. Determining the Minimum Fungistatic Concentration of Clotrimazole on Candida albicans SC5314

CTZ shows fungistatic activity towards multiple fungal species including *C. albicans* [13]. We initially analyzed the planktonic growth of *C. albicans* in the presence of CTZ, and found that the minimum fungistatic concentration was 0.3 µg/mL after a 24 h incubation at 37 °C when measuring the OD at 595 nm in the microdilution assay (Figure 1A). The maximal inhibition in growth was 80% compared to control. When measuring the ATP content in control and CTZ-treated cultures reflecting the amount of viable fungi, 0.03 µg/mL CTZ was found to significantly reduce the number of viable fungi (Figure 1B). Microscopic inspection of the fungi showed that *C. albicans* treated with 0.03 µg/mL CTZ appeared mainly in yeast form with short hyphal outgrowth, which was in sharp contrast to control fungi that have formed an extended network of long hyphae (data not shown). This observation conforms to the known inhibition of yeast-to-hypha transition by clotrimazole [17].

### 2.2. Clotrimazole-Coated Pieces of Voice Prosthesis (VPs) Resulted in Long-Term Clearance of Candida albicans on Agar Plates

VPs coated with either SRV-CTZ or placebo-varnishes were transferred daily to agar plates pre-seeded with *C. albicans* as described in the Method section, and the microbial inhibition zone was determined after 24 h. No clearance area was observed around the placebo-coated VPs, while clear inhibition zones without fungal growth was observed in the vicinity of the SRV-CTZ-coated VPs during the 60 days of testing (Figure 2A–B).

### 2.3. SRV-CTZ Coated VPs Showed Long-Term Fungistatic Effect toward Planktonic Growing C. albicans

SRV-CTZ and SRV-placebo coated VPs were tested for their long-term antifungal activity by daily transfer of the pieces to fresh planktonic growing *C. albicans*, and the relative amount of live fungi in the medium after 24 h incubation was determined by measuring the ATP content (Figure 3A). It was observed that the SRV-CTZ-coated VPs could significantly inhibit the fungal growth in the medium during the 60 days tested (Figure 3A). A residual fungal population was seen in the medium exposed to SRV-CTZ-coated VPs (Figure 3A), which is explained by the fungistatic, rather than fungicidal, activity of CTZ.

To evaluate the ability of the fungi to recover following exposure to the SRV-CTZ-coated VPs, the drop growth method was used. Three drops of the medium from each sample were seeded on agar plates, and the fungal growth was inspected after 24 h. No significant fungal growth was observed from medium exposed to SRV-CTZ-coated VPs for around 30 days (Figure 3B), indicating that the amount of CTZ released to the medium was sufficient to arrest fungi growth even for another 24 h. After 32 days of exposure to CTZ-coated VPs, small colonies of fungi could be observed on the agar plates (Figure 3B), however, these were significantly smaller than those formed after exposure to placebo-coated VPs. Even after 60 days of exposure to SRV-CTZ-coated VPs, the number of fungal colonies remained smaller than the fungal samples exposed to placebo-coated VPs (Figure 3B). There were some white spots observed on the agar from the CTZ samples up to day 5 (See Day 3 in Figure 3B), which were due to CTZ precipitates. These precipitates resulted from an initial higher CTZ release during the first days of incubation in comparison to the later time points (Figure 2B and Figure 4).

HPLC analysis of the daily released CTZ showed an initial burst effect during the first 2 days, followed by a more and less steady release kinetics for 60 days with small fluctuations (Figure 4). Most of the samples showed a CTZ concentration ranging between 2.1–2.6 µg/mL (Figure 4) which is almost 10 times higher than the MIC (Figure 1A). Even after 60 days of exposure to fluid, still 30–35% of the coating is retained on the VP. Some daily shedding of the film itself occurred when incubated with PBS. It should be noted that the sustained release assay was performed under stationary conditions which might explain the more and less similar daily release of CTZ during the period of days 7–60.

When inspecting the samples exposed to SRV-CTZ-coated VPs under a microscope, the fungi appeared mainly in yeast form with short hyphal outgrowth, instead of the prominent hyphal form observed in the samples exposed to SRV-placebo-coated VPs (Figure 5). The yeast form with short hyphal outgrowth of *C. albicans* in the samples exposed to SRV-CTZ coated VPs was observed during the entire test period of 60 days. These observations suggest that the amount of CTZ released into the medium from the SRV-coated VP was indeed sufficient for preventing both yeast to hypha transition and fungal growth.

### 2.4. CTZ-Coated VPs Prevented Biofilm Formation on the Device

It was important to study whether the SRV-CTZ coated VPs could prevent biofilm formation on the device. After 14 days exposure to planktonic growing fungi, the biofilms formed on the devices were studied by live/dead/EPS staining using a spinning disk confocal microscope (SDCM) and a high-resolution scanning electron microscope (HR-SEM). SDCM of placebo-coated VP showed classical fungal biofilm structures with a depth of 300 micron (Figure 6A–D). The hyphae are visualized by the green fluorescence of SYTO 9 (Figure 6A–B). Dead fungi are conceived by the red fluorescence of PI (Figure 6C) which is outweighed by the strong SYTO 9 staining in the merged image (Figure 6A). The EPS of the placebo-coated VP was observed in the different biofilm layers using AlexaFluor^647^-conjugated ConA (Figure 6A and D). In contrast to the placebo-coated VP, the CTZ-coated VP shows only scattered dotted SYTO 9 green fluorescence (Figure 6E–F) indicating that the fungi that have adhered to the device appear in yeast form and have been unable to form a biofilm. There was only some scarce PI and EPS staining (Figure 6G–H). Quantification of the relative fluorescence intensity of the three staining agents in CTZ-coated VP versus placebo-coated VP (Figure 7) shows a 93–98% reduction in SYTO 9-stained biomass and 99–99.5% reduction in EPS mass. Likewise, HR-SEM images after 14 days of exposure to *C. albicans* show solid hyphal biomass on placebo-coated VPs (Figure 8A–D), while a multitude of minute spherical structures with a diameter of 150–450 nm could be seen in the CTZ-coated VPs between crystals of the coated material (Figure 8E–H). These structures might be related to the coating varnish. Importantly, no fungal hyphae could be observed on the device. The HR-SEM images supported the SDCM data that fungal biofilms are formed on placebo-coated VPs, while the presence of CTZ in the film coated on VPs prevents fungal growth and biofilm formation.

## 3. Discussion

Microbial biofilm formation on medical devices, including VPs, is a major health problem resulting in replacement of the device involving additional procedures, risks, cost, and recovery [18]. Fungal biofilm formation of VP is a common cause requiring its removal, cleaning and/or installment of a new prosthesis. Fungal infections of VPs, especially during the initial period following its insertion, are a major cause of device failure, often necessitating replacement. This may also cause local or systemic inflammation. Therefore, there is a compelling need for the development of a method to inhibit the initial biofilm formation on VPs, without being reliant on patient compliance. Many strategies have been examined during the years to prevent fungal infection and biofilm formation such as coating the surfaces with antibiotics or antiseptics, or, alternatively, coating the surfaces with a material that displays low adhesive properties [18]. Despite all of these efforts, the lifespan of VPs may be limited to approximately 2–3 months. Our present approach of applying a coating on the VPs using a sustained release delivery system containing the highly potent antifungal drug CTZ holds great clinical potential. The particular advantages associated with the use of this SRV include: (1) prolonged duration of drug concentration in the targeted area; (2) controlled dosages over time; (3) low dosage per time; (4) easy application by the user; (5) no reliance on patient compliance; and (6) improved microbial and clinical outcome.

CTZ was chosen to be the drug incorporated into the SRV because of its proven antifungal properties [13]. Its low solubility in water makes it an ideal drug for the sustained release device. Another advantage is that CTZ is stable at 37 °C, obviating the concern that the drug may degrade while the VP is in place. Sustained antifungal effect was observed as the CTZ continued to be released from the SRV-coated VPs in sufficient amounts to inhibit fungal growth of *C. albicans* at least for the test period of 60 days. Significant inhibition was still observed after 60 days of exposure to *C. albicans*, implying that the protective effect of the SRV could be even longer, delaying the colonization of the VP, which at the end dictates the useful lifespan of the implant. Moreover, the amount of the drug retained in the film is sufficient to hinder biofilm formation on the prosthesis. It is likely that the antibiofilm effect of SRV-CTZ is caused by the antifungal activity of CTZ. Further studies need to be performed to determine whether CTZ has a direct antibiofilm activity. We observed that CTZ prevented yeast–hyphae transition (Figure 5), which might contribute to the antibiofilm activity since the hyphal form is the major virulence factor. The important point is that our technique allows the protection of the VPs from *Candida* biofilm formation. The dual activity of preventing the growth of fungi in the vicinity of the VPs together with the hindrance of fungal biofilm formation on the VPs for a long period of time, is crucial for the maintenance of the VPs and for improving the health of the patient. Since the throat microflora is heterogeneous, future studies should be conducted that analyze the efficacy of SRV-CTZ on mixed microflora of fungi and bacteria before proceeding to clinical trials.

Another concern is the appearance of drug-resistant fungi. There are fungal species that are intrinsic resistant to azoles [13]. However, the most prevalent fungal colonizer of VPs is the azole-sensitive *C. albicans* that is the commensal yeast found on the mucosal surfaces of the oral cavity and gastrointestinal tract [2]. Long-term treatment with azole drugs has not usually led to drug resistance (The SCCNFP/0706/03 Report), since the resistance is not transferred between pathogenic fungi and the mechanisms of resistance differ from those seen in bacteria. In our experimental design, the daily transfer of SRV-CTZ coated VP pieces from an old *C. albicans* culture to a new *C. albicans* culture resulting in passive transfer of the old cultures to the new one, did not lead to the appearance of CTZ-resistant fungi even after 60 days. Importantly, our study is a proof of principle that incorporating an antifungal drug into a sustained release varnish has the potential to prevent *C. albicans* biofilm formation and thus is expected to prolong the lifetime of the device. Further studies should focus on incorporating additional drugs to broaden both the spectrum of antifungal activity as well as antibacterial activity.

The results of this study suggest that SRV-CTZ-coated VPs offers a flexible platform for designing coatings to protect VP surfaces from fungal infection and biofilm formation. Pharmaceutical formulations similar to the proposed SRVs of this type have been used in clinical trials in the oral cavity of human and in catheter-associated urinary tract infections in dogs [10,11,12]. In these studies, the active component of the SRV was either chlorhexidine or thiazolidinedione-8. More so, similar SRV-CTZ preparations have been clinically tested as a pharmaceutical indication for oral candidiasis, supplementing the traditional local therapy [16,19]. In the study of Czerninski et al. [19], the SRV-CTZ was applied on the buccal side of the oral cavity, and the CTZ concentration in the saliva was found to be above the minimum inhibitory concentration (MIC) during the test period of 5 h. In a subsequent study [16], SRV-CTZ was applied on denture stomatitis patients and compared to CTZ troches (Oralten). After 14 days, the SRV-CTZ treated patients showed significant lower levels of *Candida* on the denture surfaces and in saliva, and had a better compliance to the medication [16]. These studies relied on sufficient release of CTZ to the surroundings to suppress *Candida* infection. These studies also showed that the SRV-CTZ was well tolerated. In the present study, we utilized a similar sustained release varnish to show that coating VPs with SRV-CTZ could also prevent *C. albicans* biofilm formation on the device for a long time period, besides providing a protective surrounding. Given the current platform, the recommendation for preventive therapy with daily antifungal mouthwashes might become redundant and needs re-evaluation. Furthermore, CTZ-coating of voice prosthesis is expected to potentially improve the treatment outcomes as it eliminates the factor of patient compliance.

## 4. Conclusions

Our data establish a proof of principle that coating VPs with a sustained release film containing the antifungal drug CTZ both prevents biofilm formation on the VPs and provides long-term protection of the surrounding environment against fungal infections. This strategy should significantly prolong the lifespan of voice prostheses and improve clinical outcomes in patients.

## 5. Materials and Methods

### 5.1. Voice Prosthesis

The Blom-Singer classic indwelling VPs (InHealth Technology, Carpinteria, CA, USA) were cut into 5–6 pieces of similar sizes and sterilized by immersing them in 70% alcohol overnight, and then dried aseptically.

### 5.2. Clotrimazole and Placebo Varnishes

The SRV-CTZ was composed of 1.2 g Clotrimazole (CTZ, Sigma), 0.9 g ethylcellulose (Ashland Specialty Ingredients, Wilmington, USA), 0.9 g Klucel EF (Ashland Specialty Ingredients, Switzerland), and 12 mL ethanol [16]. The dry film contained 40% (*w*/*w*) CTZ. The placebo varnish contained 0.9 g ethyl cellulose, 0.9 g Klucel EF, and 12 mL ethanol.

### 5.3. Coating of Voice Prosthesis Pieces

Sterile VP pieces were coated by immersing them in the SRV and drying them at room temperature to allow for the formation of a film on the surface of each piece. This process was repeated twice. The pieces were allowed to dry completely for 3 days before use. The average SRV coating of the pieces was 10–12 mg containing 4–4.8 mg of CTZ. The resulting film adhered well to the VPs and remained attached to the material throughout the tested period of 60 days.

### 5.4. Fungi Culture Conditions

*Candida albicans* SC5314 (ATCC MYA-2876) from a frozen vial was seeded on potato dextrose agar (PDA) (Acumedia, Neogen, MI, USA) plates and incubated at room temperature. *C. albicans* colonies were picked up daily from the agar plates and suspended in RPMI-1640 medium (Sigma, Ronkonkoma, NY USA), and used for the assays described below.

### 5.5. Determining the Minimum Fungistatic Concentration of Clotrimazole in Microdilution Assay

Two hundred microliters of a suspension of *C. albicans* in RPMI with an optical density (OD) of 0.3 at 600 nm were seeded in 96 flat-bottom transparent wells (Corning) in the presence of increasing concentrations of CTZ or an equivalent concentration of ethanol that served as the control. The OD at 600 nm was measured after a 24 h incubation in Infinite M200PRO plate reader (Tecan, Trading AG, Männedorf, Switzerland) [20]. The percentage of fungi in the samples was calculated using the following formula: (OD_CTZ_/OD_PL_) × 100, where OD_CTZ_ is the average OD_600 nm_ of the CTZ-treated samples and OD_PL_ the average OD_600 nm_ of the control samples.

### 5.6. Agar-Based Activity Assay

PDA plates were seeded daily with 200 µL of freshly prepared *C. albicans* suspension at an OD_600 nm_ of 0.3. Two pieces of CTZ-coated VPs and two pieces of placebo-coated VPs were transferred daily to *C. albicans*-coated agar plates that were incubated at 37 °C for 24 h. The clearance zone was calculated by [(d1/2) × (d2/2) × π] − [(d3/2)2 × π], where d1 and d2 are the two diameters of the clearance zone and d3 is the diameter of the VPs.

### 5.7. Planktonic-Based Activity Assay

Five pieces of CTZ-coated VPs and five pieces of placebo-coated VPs were transferred daily to 1 mL freshly prepared *C. albicans* suspension at an OD_600 nm_ of 0.025 in 48-flat-bottomed tissue culture wells (Corning) and incubated at 37 °C for 24 h, except for the weekends where the incubation endured for 48 h. Following each incubation period, the coated VPs were transferred from the old culture to a fresh culture. This was repeated for 14 or 60 days. One piece of each kind (treated and placebo) of coated SRV-VPs was taken for spinning disk confocal microscopy (SDCM) and high-resolution scanning electron microscopy (HR-SEM) for studying biofilm formation (see below) after 14 days of exposure to *C. albicans*. The other VP pieces were used for continued daily incubation with fresh *C. albicans* suspension for 60 days. Daily analysis of the *C. albicans* load in the wells were performed using the drop agar plate method and by measuring the relative ATP content using the BacTiter^Glo^ microbial cell viability assay (Promega Corporation, Madison, WI, USA) (see below), where the ATP content in the placebo group was set to 100%.

### 5.8. Drop Agar Plate Method

To study the ability of the *C. albicans* to grow from the samples exposed to SRV-CTZ or SRV-placebo-coated VPs in the planktonic-based activity assay, the fungi in the wells were suspended to a homogenous suspension, and three drops of 10 µL were plated from each well on PDA plates, that were incubated at 37 °C for 24 h.

### 5.9. BacTiterGlo Microbial Cell Viability Assay

To determine the relative amount of live *C. albicans* in the media of the planktonic-based activity assay, 100 µL of the homogeneous fungal suspension from each well were transferred to white 96-well flat-bottomed microplates, to which 100 µL of the BacTiter^Glo^ reagent (Promega Corporation, Madison, WI, USA) was added [21]. After 20 min of shaking at room temperature, the luminescence was measured in an Infinite M200PRO plate reader (Tecan, Männedorf, Switzerland). The percentage of viable fungi in the SRV-CTZ group in comparison to SRV-placebo group was calculated using the following formula: (Lum_CTZ_/Lum_PL_) × 100, where Lum_CTZ_ is the average luminescence of the samples from the SRV-CTZ group and Lum_PL_ the average luminescence of the samples from the SRV-placebo group.

### 5.10. Determination of the Daily Release of Clotrimazole from SRV-CTZ-coated VPs

SRV-CTZ-coated VP was daily incubated in 1 mL sterile PBS for 24 h at 37 °C for a period of 60 days. The daily release of CTZ into PBS was determined by high-performance liquid chromatography (HPLC) according to the method of de Bruijn et al. [22]. Next, 100 µL of each sample were automatically injected into a HPLC column packed with Inertsil ODS-80A (5µm particle size; 150 mm (length) × 4.6 mm (internal diameter); GL Science, Tokyo, Japan) using the HP 1090 series HPLC system (Hewlett Packard, Palo Alto, CA, USA). The mobile phase was composed of water–acetonitrile–tetrahydrofuran–ammonium hydroxide–trimethylamine at a ratio of 45:50.2:2.5:0.1:0.1 (*v*/*v*), pH = 6.0. The column effluent was monitored by UV at a wavelength of 206 nm, and the clotrimazole concentration in the samples were determined according to a standard curve made from known clotrimazole concentrations.

### 5.11. Live/Dead SYTO 9/Propidium Iodide (PI) Staining and EPS Staining of VPs

After 14 days of exposure to *C. albicans* in the planktonic-based activity assay, the SRV-VPs were washed twice in sterile PBS, and stained with 3.25 µM SYTO 9 green fluorescent nucleic acid stain (Invitrogen, Life Technologies Corporation, Eugene, OR, USA), 2.5 µg/mL propidium iodide (PI; Sigma, St. Louis, MO, USA) and 20 µg/mL AlexaFluor^647^-conjugated ConA (Invitrogen, Life Technologies Corporation, Eugene, OR, USA) for 20 min at room temperature in the dark [20]. Thereafter, the samples were washed twice in PBS, fixed in 4% paraformaldehyde in PBS for 1 h, and kept in 50% glycerol in DDW until visualizing the biofilms under a spinning disk confocal microscope (Nikon Yokogawa W1 Spinning Disk, Tokyo, Japan, with 50 µm pinholes; SDCM). The biofilm depth was assessed by capturing optical cross-sections at 3 μm intervals from the bottom of the biofilm to its top. The SYTO 9 green fluorescence dye, which enters both live and dead bacteria, was visualized using 488 nm excitation and 515 nm emission filters. The PI red fluorescence dye, which only penetrates dead bacteria, was measured using 543 nm excitation and 570 nm emission filters. AlexaFluor^647^-ConA staining of EPS was measured using 638 nm excitation wavelength and 680 nm emission filter. Thus, live bacteria fluoresce green light, while dead bacteria fluoresce both green and red light. Three-dimensional images of the formed biofilms were reconstructed using the NIS-Element AR software. This software was also used to analyze the fluorescence intensity of SYTO 9, PI and EPS staining in each captured layers of the biofilms. The staining on SRV-CTZ-coated VP was compared to that of SRV-placebo-coated VP.

### 5.12. HR-SEM Imaging of VPs

After 14 days of exposure to *C. albicans* in the planktonic-based activity assay, the VPs were washed twice in sterile DDW, and fixed in 4% paraformaldehyde in DDW for 1 h. Then the samples were washed again in DDW, dried, and coated with iridium and visualized using a FEI Magellan 400 L High-Resolution Scanning Electron Microscope (HR-SEM; FEI Company, Hillsboro, OR, USA) at 200×–10,000× magnification.

### 5.13. Statistical Analysis

The data are expressed as the average ± standard deviation. The number of coated pieces used in each experiment varied from 2–5. Statistical analysis was performed using the Microsoft excel software. Student's *t*-test was used to compare clotrimazole-coated VPs with placebo-coated VPs, with a P value less than 0.05 considered significant.

## Figures and Tables

**Figure 1 molecules-26-05395-f001:**
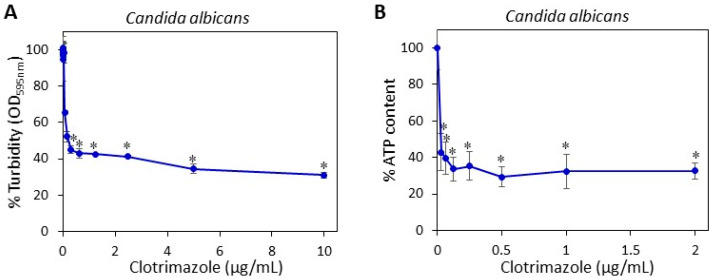
The antifungal effect of CTZ on *C. albicans*. (**A**) *C. albicans* at an initial OD_600 nm_ of 0.3 was exposed to increasing concentrations of CTZ, and the OD_595 nm_ after 24 h was measured in a Tecan M200 plate reader; (**B**) *C. albicans* at an initial OD_600 nm_ of 0.3 was exposed to increasing concentrations of CTZ for 24 h, and the ATP content was measured by luminescence in a Tecan M200 plate reader. * *p* < 0.05 versus control. N = 3.

**Figure 2 molecules-26-05395-f002:**
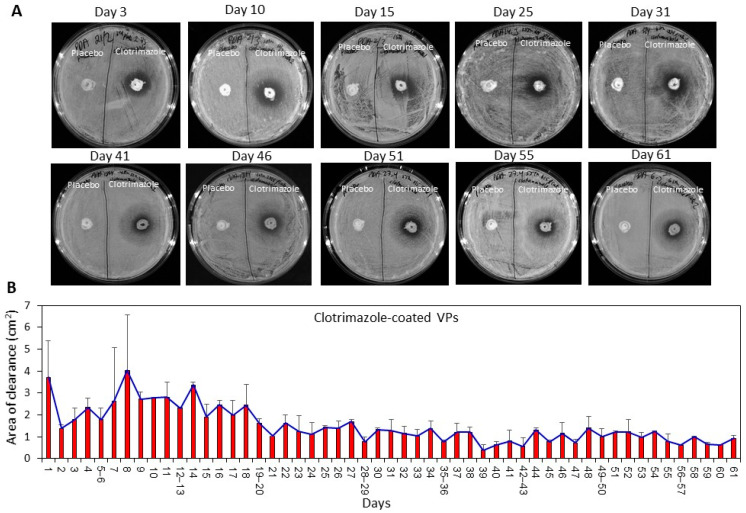
SRV-CTZ-coated VPs prevented fungal growth in its vicinity for long periods of time. (**A**,**B**) SRV-CTZ- and placebo-coated VPs were daily placed on *C. albicans*-coated agar plates (**A**), and the clearance area determined (**B**). The placebo-coated VPs did not cause any clearance of the fungi. N = 2.

**Figure 3 molecules-26-05395-f003:**
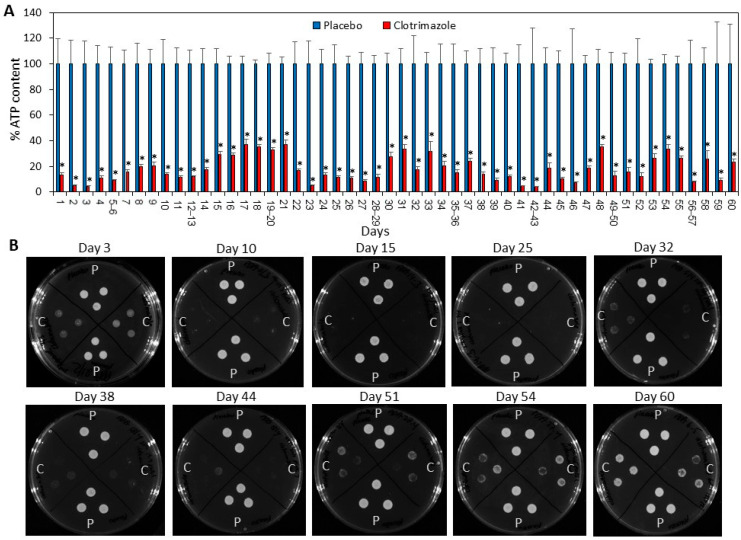
SRV-CTZ-coated VPs prevented the planktonic growth of *C. albicans*. (**A**) SRV-CTZ- and placebo-coated VPs were daily incubated with fresh *C. albicans* cultures, and the relative ATP content of the fungi after a 24 h incubation was determined. * *p* < 0.05 versus placebo. (**B**) 3 drops of 10 µL from each sample from experiment described in A, were seeded on agar plates to analyze the ability of the fungi to grow after being exposed to SRV-CTZ- or placebo-coated VPs. C = Samples exposed to SRV-CTZ-coated VPs. P = samples exposed to SRV-placebo-coated VPs. N = 3–5.

**Figure 4 molecules-26-05395-f004:**
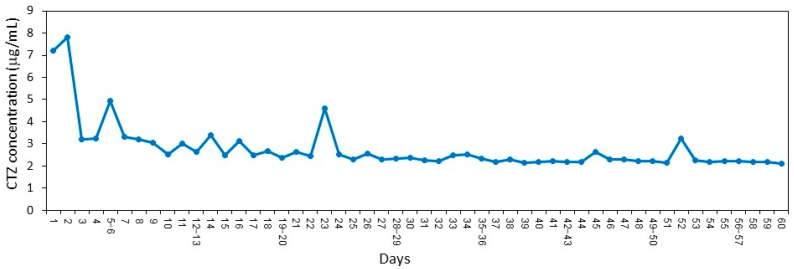
HPLC analysis of the daily release of CTZ from VP coated with SRV-CTZ. VP coated with SRV-CTZ was daily incubated in 1 mL PBS for 24 h at 37 °C and the CTZ concentration of the soluble fraction was determined by HPLC according to a standard curve using known concentrations of CTZ in PBS. N = 1.

**Figure 5 molecules-26-05395-f005:**
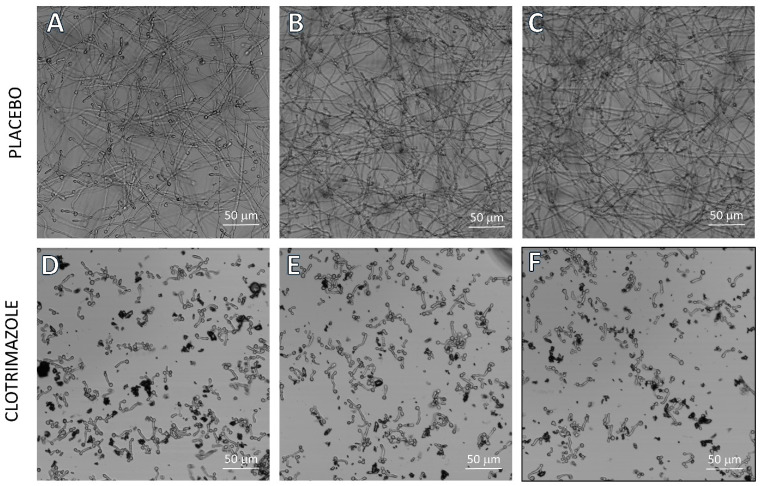
Microscope images of *C. albicans* exposed to SRV-CTZ- and placebo-coated VPs. SRV-CTZ- and placebo-coated VPs were daily exposed to fresh *C. albicans* cultures, and at day 36, images of three different samples were captured using a confocal microscope (Nikon Eclipse Ti-U). The fungi exposed to SRV-CTZ-coated VPs appear in yeast form with short hyphal outgrowth, in contrast to those exposed to SRV-placebo-coated VPs, which form networks of long hyphae. (**A**–**C**) Placebo-coated VPs. (**D**–**F**) SRV-CTZ-coated VPs. N = 3.

**Figure 6 molecules-26-05395-f006:**
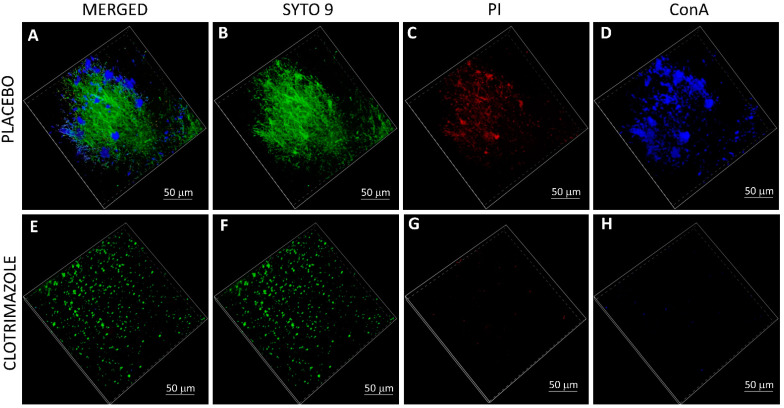
Scanning disk confocal microscopy (SDCM) of SRV-CTZ- versus placebo-coated VPs. SRV-CTZ- and placebo-coated VPs that have been daily incubated with fresh *C. albicans* cultures for 14 days were washed and stained with SYTO 9 (green fluorescence) (**B**,**F**), propidium iodide (PI, red fluorescence) (**C**,**G**) and AlexaFluor647-conjugated ConA (blue color) (**D**,**H**). The stained samples were visualized by a Nikon spinning disk confocal microscope. (**A**,**E**) are the merged images of the three staining dyes. (**A**–**D**) Placebo-coated VPs. (**E**–**H**) SRV-CTZ-coated VPs. N = 1.

**Figure 7 molecules-26-05395-f007:**
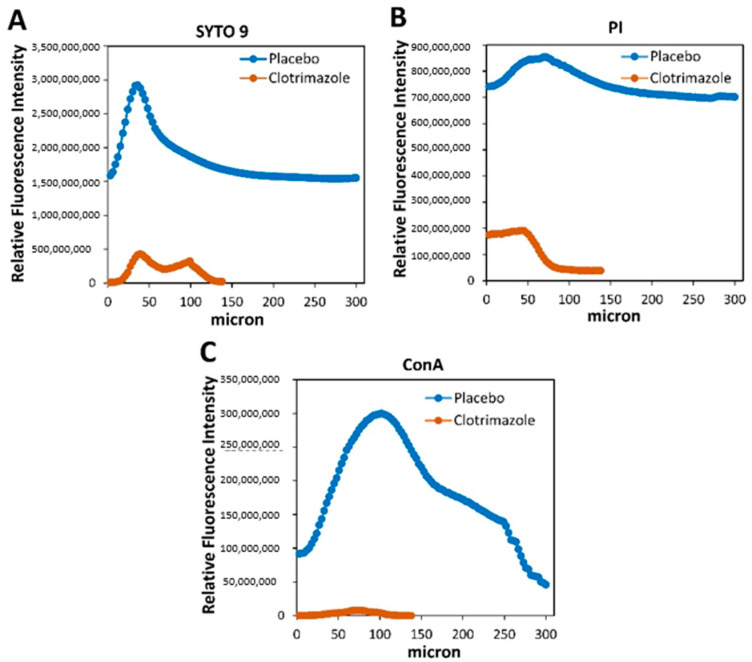
Quantification of the relative fluorescence intensities of SRV-CTZ- versus placebo-SRV-coated VPs analyzed on SDCM. The relative fluorescence intensities of triplicate images captured on spinning disk confocal microscope were analyzed using the NIS-Element AR software. (**A**) SYTO 9 staining. (**B**) PI staining. (**C**) AlexaFluor^647^-conjugated ConA staining. N = 1.

**Figure 8 molecules-26-05395-f008:**
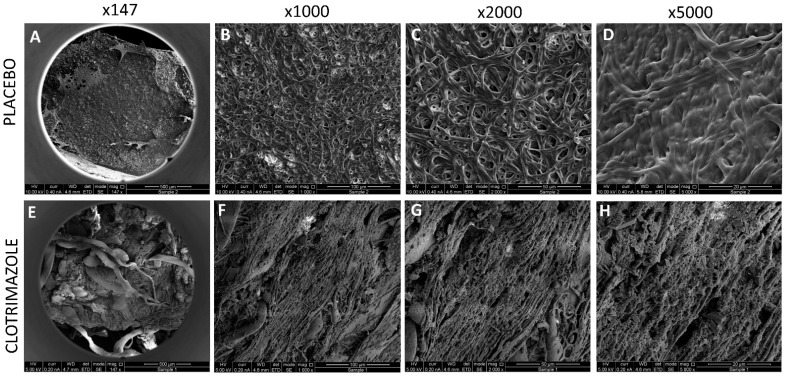
HR-SEM images of SRV-CTZ- versus placebo-coated VPs. SRV-CTZ- and placebo-coated VPs that have been daily incubated with fresh *C. albicans* cultures for 14 days were washed, fixed, coated with iridium and visualized using a FEI Magellan 400 L High-Resolution Scanning Electron Microscope. (**A**–**D**) Placebo-coated VPs. (**E**–**H**) SRV-CTZ-coated VPs. N = 1.

## Data Availability

Raw data are available upon reasonable request.

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
