# Peer review of "Voice Prosthesis Coated with Sustained Release Varnish Containing Clotrimazole Shows Long-Term Protection against Candida albicans: An In Vitro Study"

_molecules, 2021, doi:10.3390/molecules26175395_

Round 1
Reviewer 1 Report
Dear Editor
I have reviewed the manuscript entitled "Voice Prosthesis Coated with Sustained Release Varnish 2 Containing Clotrimazole Shows Long-Term Protection against Candida albicans". This manuscript presents relevant information about the potential use of coating voice prosthesis with sustained-release varnish with clotrimazole to prevent fungal infections and biofilm formation in voice prosthesis. This work is interesting, well presented, and the results are adequately described. However, I suggest some changes before being accepted for publication.
- I suggest identifying in the title if the study reports in vitro
- In the introduction section, the authors mentioned that other approaches had been used to prevent fungal contamination. However, it is not clear why these methods have not worked. I suggest authors explain the flaws of previous experiments and why their proposal could improve those flaws.
- I suggest changing the keywords for words that are not in the title but are pertinent to the study to increase the search options.
- The authors indicated that formed biofilms are extremely difficult to eradicate due to their less sensitivity to antimicrobials. Taking this into account, the authors proposed to prevent biofilm formation and demonstrated that the coated voice prosthesis inhibited biofilm formation. However, I think that the antibiofilm effect is obtained by reducing cell viability instead of process inhibition involved in biofilm formation. I suggest that the authors carry out some tests to determine if the effect was on inhibiting biofilm formation, such as motility, EPS production, and gene expression. Otherwise, I suggest clarifying this point in the manuscript.
- Are there reports of resistance to clotrimazole? Wouldn't resistance develop if this prosthesis is worn for 60 days?
- In the discussion section, the authors mentioned that there are already clinical trials using similar formulations. What would be the novelty of this study?
- In the material and methods section, it is unclear when the authors describe that SRV-CTZ and SRV-placebo coated VP are daily transferred to fresh planktonic growing cells. Are these VPs in previous planktonic cultures, or are they recently coated? From where are they transferred? It is not clear how they were daily incubated and evaluated for 14 or 60 days? I can imagine that the same piece of VP is transferred from old culture to a fresh culture each day for 14 days. I suggest to re-write those sentences to have a more precise methodology.
- Why was the biofilm formation evaluated for 14 days?
- In section 5.6, lines 326-327, "The others were used for continued daily incubation with fresh C. albicans suspension." I suggest indicating how long it was incubated.
- I suggest including in the statistical analysis how many replications and repetitions were made of the experiments

Author Response
We want to thank the reviewer for the good comments.
Please find below point-to point answers to the queries.
I have reviewed the manuscript entitled "Voice Prosthesis Coated with Sustained Release Varnish 2 Containing Clotrimazole Shows Long-Term Protection against Candida albicans". This manuscript presents relevant information about the potential use of coating voice prosthesis with sustained-release varnish with clotrimazole to prevent fungal infections and biofilm formation in voice prosthesis. This work is interesting, well presented, and the results are adequately described. However, I suggest some changes before being accepted for publication.
- I suggest identifying in the title if the study reports in vitro
We have added "An in vitro study" to the title.
- In the introduction section, the authors mentioned that other approaches had been used to prevent fungal contamination. However, it is not clear why these methods have not worked. I suggest authors explain the flaws of previous experiments and why their proposal could improve those flaws.
We have now added text to the Introduction describing the results of the documented approaches. We have developed a different method, which is based on the concept of sustained release delivery technology of the active agent, which is unique. It allows the drug to be released over time which is of a great pharmacological advantage.
- I suggest changing the keywords for words that are not in the title but are pertinent to the study to increase the search options.
We have now added the following keywords: Laryngeal malignancies, Laryngectomy
- The authors indicated that formed biofilms are extremely difficult to eradicate due to their less sensitivity to antimicrobials. Taking this into account, the authors proposed to prevent biofilm formation and demonstrated that the coated voice prosthesis inhibited biofilm formation. However, I think that the antibiofilm effect is obtained by reducing cell viability instead of process inhibition involved in biofilm formation. I suggest that the authors carry out some tests to determine if the effect was on inhibiting biofilm formation, such as motility, EPS production, and gene expression. Otherwise, I suggest clarifying this point in the manuscript.
The aim of our study was not to determine the mechanism of action of clotrimazole, which is a broad-acting anti-fungal drug of the azole family that has been used for treating fungal infections for decades. Rather, our aim was to study the ability of clotrimazole when incorporated into a sustained release varnish applied on VPs, to prevent Candida biofilm formation on the device. Indeed, in this study we proved that CTZ-SRV coating of VPs can prevent biofilm formation of Candida albicans on the device as well as exerting an anti-fungal effect around the coated VPs, using different methods. The SDCM images show that CTZ-SRV caused a strong reduction in both biofilm mass and EPS production. We agree with the reviewer that the anti-biofilm effect of CTZ-SRV might be indirectly caused by its anti-fungal activity. In order to clarify this issue, we have added the following text to Discussion: "It is likely that the anti-biofilm effect of CTZ-SRV is caused by the anti-fungal activity of CTZ. Further studies need to be performed to determine whether CTZ has a direct anti-biofilm activity. We observed that CTZ prevented yeast-hyphae transition (Figure 5), which might contribute to the anti-biofilm activity since the hyphal form is the major virulence factor. The important point is that our technique allows the protection of the VPs from Candida biofilm formation."
- Are there reports of resistance to clotrimazole? Wouldn't resistance develop if this prosthesis is worn for 60 days?
We have now added the following text to Discussion to clarify this issue: "Another concern is the appearance of drug-resistant fungi. There are fungal species that are intrinsic resistant to azoles [13]. However, the most prevalent fungal colonizer of VPs is the azole-sensitive albicans that is the commensal yeast found on the mucosal surfaces of the oral cavity and gastrointestinal tract [2]. Long-term treatment with azole drugs has usually not led to drug resistance (The SCCNFP/0706/03 Report), since the resistance is not transferred between pathogenic fungi and the mechanisms of resistance differ from those seen in bacteria. In our experimental design, the daily transfer of CTZ-SRV coated VP pieces from an old C. albicans culture to a new C. albicans culture resulting in passive transfer of the old cultures to the new one, didn't lead to the appearance of CTZ-resistant fungi even after 60 days. Importantly, our study is a proof-of-principle that incorporating an anti-fungal drug into a sustained release varnish has the potential to prevent C. albicans biofilm formation and thus is expected to prolong the lifetime of the device. Further studies should focus on incorporating additional drugs to broaden both the spectrum of anti-fungal activity as well as anti-bacterial activity."
- In the discussion section, the authors mentioned that there are already clinical trials using similar formulations. What would be the novelty of this study? The previous clinical trials using SRV-CTZ were a proof of concept in the oral cavity only. These studies showed that clotrimazole in a sustained release varnish is safe and can reduce the Candida levels in the oral cavity. In the present study, we have taken the technology a step forward to design a novel application that can prevent albicans biofilms on VP. The surface on which the SRV-CTZ is coated is totally different from what has been tested before, and the assays examining the efficacy of the SRV are different too. In order to clarify this issue, we have added the following text to the Discussion: "In the study of Czerninski et al. [19], the SRV-CTZ was applied on the buccal side of the oral cavity on the buccal mucosa, and the CTZ concentration in the saliva was found to be above the MIC during the test period of 5 h. In a subsequent study [16], SRV-CTZ was applied on denture stomatitis patients and compared to CTZ troches (Oralten). After 14 days, the SRV-CTZ treated patients showed significant lower levels of Candida on the denture surfaces and in saliva, and had a better compliance to the medication [16]. These studies relied on sufficient release of CTZ to the surroundings to suppress Candida infection. These studies also showed that the SRV-CTZ was well tolerated. In the present study, we utilized a similar sustained release varnish to show that coating VPs with SRV-CTZ could also prevent C. albicans biofilm formation on the device for a long time period, besides providing a protective surrounding."
- In the material and methods section, it is unclear when the authors describe that SRV-CTZ and SRV-placebo coated VP are daily transferred to fresh planktonic growing cells. Are these VPs in previous planktonic cultures, or are they recently coated? From where are they transferred? It is not clear how they were daily incubated and evaluated for 14 or 60 days? I can imagine that the same piece of VP is transferred from old culture to a fresh culture each day for 14 days. I suggest to re-write those sentences to have a more precise methodology. We have accordingly added the following sentence to the Method section: "Following each incubation period, the coated VPs were transferred from the old culture to a fresh culture. This was repeated for 14 or 60 days."
- Why was the biofilm formation evaluated for 14 days?
The extent of biofilm formation on the coated VPs was evaluated after 14 times of exposure to fresh albicans cultures. Due to limited VP material, we decided to take samples for SDCM and HR-SEM at one time point, and found that 14 times of exposure to the fungi is a reasonable time period.
- In section 5.6, lines 326-327, "The others were used for continued daily incubation with fresh C. albicans suspension." I suggest indicating how long it was incubated.
We have accordingly changed the sentence to: " The other VP pieces were used for continued daily incubation with fresh albicans suspension for 60 days."
- I suggest including in the statistical analysis how many replications and repetitions were made of the experiments
We have added the N values in the figure legends and added the following sentence to Section 5.13: "The number of coated pieces used in each experiment varied from 2-5."

Reviewer 2 Report
Very interesting paper, showed crucial for future used
voice prosthesis coated with Sustained Release Varnish Containing Clotrimazole. Nevertheless, few details need explenation. Below point by point I try to show main concerns. 1. Authors on Fig. 1 shows viability. But they not determine viability test. Shows onlu optical density is not enough. Authors in this maner shows only that one culture had decresed optical density compared to control. In Candida species is important to show cells under microskope. Therefore please determine viability by e.g. iodium propidium. It is posible that cells are live but have decresed metabolism. In this situation your resault may not be in consis with OD.
Why thay do not calculate numer of cells during ATP test? Authors compare culture with different numer of cells? I think that is not apropriate approche.
Minor:
Please cerfully standarise unit
Author Response
We want to thank the reviewer for the good comments.
Please find below point-to point answers to the queries.
- Very interesting paper, showed crucial for future used voice prosthesis coated with Sustained Release Varnish Containing Clotrimazole. Nevertheless, few details need explenation. Below point by point I try to show main concerns. 1. Authors on Fig. 1 shows viability. But they not determine viability test. Shows onlu optical density is not enough. Authors in this maner shows only that one culture had decresed optical density compared to control. In Candida species is important to show cells under microskope. Therefore please determine viability by e.g. iodium propidium. It is posible that cells are live but have decresed metabolism. In this situation your resault may not be in consis with OD.
We have accordingly changed the title of the y-axis in Figure 1 from % Viability to % Turbidity. Figure 1 is presented to show the minimum fungistatic concentration of clotrimazole, a drug that is well-known to have fungistatic activity. Since the drug is fungistatic, the propidium iodide staining cannot be used in this setting. The ATP assay complements the OD reads, showing strong reduction in the metabolic activity in response to clotrimazole. This corresponds to the reduced number of fungi seen under the microscope. Since the effect of clotrimazole on albicans can be monitored by using the BacTiterGlo viability assay (Figure 1B), this assay is reliable for measuring the anti-fungal effect of CTZ released from SRV-CTZ-coated VPs.
- Why thay do not calculate numer of cells during ATP test? Authors compare culture with different numer of cells? I think that is not apropriate approche.
The experiments were performed at 37°C resulting in yeast-to-hyphae transition in the control samples where a network of hyphae is formed (Confer Figure 5). This complicates the counting of cells. The BacTiterGlo ATP assay was found to be a sensitive assay to measure the response of albicans to clotrimazole (Figure 1B), and was therefore used to compare the effect of SRV-CTZ versus Placebo-SRV coated VPs on C. albicans growth (Figure 3A). The ATP assay was complemented by the drop agar method (Figure 3B). Both assays show that SRV-CTZ coated VPs reduces the number of fungi in comparison to Placebo-SRV coated VPs. This was also shown by microscopic inspection (Figure 5).
The cultures exposed to SRV-CTZ coated VPs do show a lower number of fungi that those exposed to Placebo-SRV which is the whole aim of using the SRV-CTZ coating. The ATP assay was used to quantify this difference.
- Minor:
Please cerfully standarise unit
All the figures are presented as the effect of SRV-CTZ-coated VP pieces in comparison to Placebo-coated VP pieces. We have not used "units" in our manuscript.

Round 2
Reviewer 2 Report
Well done!